# The Economic Case for Electric Vehicles in Public Sector Fleets: An Italian Case Study

**Romeo Danielis** [1,*]**, Mariangela Scorrano** [1]**, Marco Giansoldati** [1]  **and Stefano Alessandrini** [2]

[1] Department of Economics, Business, Mathematics and Statistics "Bruno de Finetti" (DEAMS), Trieste University, 34127 Trieste, Italy; mscorrano@units.it (M.S.); mgiansoldati@units.it (M.G.)
[2] Area di Ricerca Scientifica e Tecnologica di Trieste—Area Science Park, 34149 Trieste, Italy; stefano.alessandrini@areasciencepark.it
[*] Correspondence: romeo.danielis@deams.units.it

**Abstract:** The paper investigates whether it makes economic sense to use electric vehicles (EVs) in the public sector fleet. Thanks to the data collected in 2018 in 77 public sector entities in an Italian region, Friuli Venezia Giulia, we compare the total cost of ownership of a battery electric vehicle with that of a similar internal combustion engine one. We provide estimates for four scenarios (status quo, social cost internalization, price discounts and a combination of the last two) for three groups of public entities (local health authorities, municipalities and special purpose authorities) regarding passenger cars and mixed-use small light commercial vehicles. We find that, with the current price and cost structure, it makes economic sense to adopt EVs for a positive although relatively small percentage of the public sector fleet.

**Keywords:** public sector fleet; electric vehicle; total cost of ownership

## 1. Introduction

The public sector owns and manages a considerable number of vehicles and equipment. Public fleet managers are required to meet several, often conflicting objectives such as adhering to severe budget constraints (especially when public finances are tight), catching up on vehicle replacements, adding management technology to enhance fleet operations and greening their fleets to reduce local air pollution, noise pollution and carbon dioxide emissions.

Electric vehicles (EVs) represent an interesting option to realize some of these goals. While it is commonly accepted that EVs contribute to reduce local air emissions, it is far from clear whether their acquisition is justified from an economic point of view. Moreover, EVs need to be operationally suitable for the task that they are required to perform. Fleets managers, hence, have the complex responsibility to take into account the nature of use of the vehicles, their daily distances and usage schedule, the origin and destination of the trips they make and their charging requirements.

Various public sector bodies around the world have already made headway in this direction. In Italy, the Friuli Venezia Giulia (FVG) Region embraced this agenda in 2017, within the H2020 European project Noemix [1], whose goals are reducing $CO_2$ emissions and air pollution, increasing the share of renewables in the electricity mix, improving overall energy efficiency and substituting the current vehicles with electric ones. With the Noemix project, the authors of this paper had the chance to collect data on the public sector fleets in operation in several agencies and authorities. The data provided us with information on the number and model type of vehicles, their powertrain and fuel-use, their age and annual distance traveled.

In this paper, we report on some of these results and perform a total cost of ownership (TCO) analysis to answer the question of whether it makes economic sense to substitute these internal

combustion engine vehicles (ICEVs) with battery electric vehicles (BEVs, Through the paper, we will not consider plug-in hybrid electric vehicles (PHEVs) for two reasons: 1) they are not part of the Noemix project, and b) their electric-only distance could be limited, hence, they do not represent a full paradigm change relative to the current ICE technology. However, we acknowledge that they can represent a transitory pathway towards the more challenging electric-only technology). We grouped the regional public sector entities by type of service offered, distinguished among passenger cars and mixed-use light commercial vehicles (LCVs), estimating which percentage of their current ICE fleet is economically justified to substitute under four scenario assumptions, given the 2020 price structure prevailing in Italy.

As illustrated in the next section, to the best of our knowledge this is one of the few applied research papers on this topic, certainly the first one concerning Italy.

The main contributions to the existing literature of this paper are the following:

- presenting and analyzing the main characteristics of the public sector fleets for 77 public sector entities in the Friuli Venezia Giulia Region, Italy, in terms of fleet size and composition, age distribution, fuel and model type and annual distance traveled. Although very case specific, there might be similarities with other Italian or non-Italian public sector entities;
- developing and applying a TCO model to compare the cost competitiveness of alternative propulsion systems in the medium-to-small passenger car and LCV segments under alternative scenarios: a) a status quo scenario with 2020 prices; b) a social cost internalizing scenario; c) a discounted-purchase price Scenario; d) and a combination of b) and c);
- estimating, under the above scenario assumptions, the break-even annual traveled distance that makes an EV competitive with respect to an ICE counterpart. Consequently, we are able to estimate the percentage of the public sector fleets that makes economic sense to be electric.

Although this paper deals with only one of the various aspects relevant for public fleet procurement decisions (the economic case for EVs), the findings of the paper might help public fleet managers to make economically informed decisions to be shared with the policy makers and the taxpayers.

The paper is organized as follows. Section 2 reviews the related literature, Section 3 describes the main characteristics of the vehicular fleets of the Public Administration in the Friuli Venezia Giulia Region, Section 4 illustrates the Total Cost of Ownership model and the results of its implementation, Section 5 reports on the vehicles considered and on the parameters used to estimate the model. Section 6 discusses the main results and develops scenario analyses. Section 7 draws some conclusions.

## 2. Related Literature

In this paper, we apply the TCO methodology to evaluate the economic case for EVs adoption in the FVG Region public sector fleets. The TCO methodology is widely adopted to compare different products. It represents a tool to estimate the true financial cost of a good [2,3]. However, as explained by [4], TCO models might have two cost components: one including the costs borne by the vehicle user (the consumer-oriented TCO), and one including the costs borne by society, as for example air and noise pollution (the society-oriented TCO). This paper will use both of them. Although estimating a TCO model presents computational challenges, it provides a useful information to consumers, fleet managers, manufacturers and policy makers.

One of the main difficulty is the uncertainty connected to the future stream of costs [5]. There are three types of uncertainty: technical, economic and political. Since BEVs are a relatively new technology, subject to continuous technological improvements, a major technical uncertainty concerns battery degradation, with implications on the substitution costs and the vehicle's resale value. Furthermore, BEV efficiency in real traffic, at different speeds and in different weather conditions, as well as the actual maintenance and repair costs are uncertain. As more experience with BEVs is gained, technical uncertainty reduces. Economic uncertainty is related to future fuel and energy prices [6,7]. Political uncertainty is associated with traffic regulation (tolls, access restrictions, parking tariffs) and incentives'

mechanisms (e.g., purchase subsidies and circulation tax exemptions) that could affect the relative cost advantages of the different propulsion systems. In the existing TCO literature, the uncertainty aspect has been incorporated using probabilistic TCO models ([8,9]). The economic uncertainty connected with future prices has been studied by [10] using an assortment of future cost scenarios, or by [11] combining tools from the TCO and technology selection literature.

A second important aspect to consider is that the TCO is inherently vehicle-, region- and individual-specific. Cost competitiveness among propulsion systems depends on the vehicle market segment (small, medium or large cars, SUVs, LCVs, trucks, etc.). Up to now, car manufacturers have focused on the more profitable luxury segment of the car market. Only some manufactures ventured to supply electric medium- to small passenger cars or LCVs. Country and regional specificities are connected with policy choices regarding vehicles' and fuel taxation. Finally, TCO is individual-specific related to the driving style, traveling and charging habits/needs and vehicle use intensity (measured by the average annual distance traveled). Several papers have explored these TCO differences. The authors of this study have performed previous TCO applications to different market segments including private cars ([9,12–14]), LCVs ([15]) and taxis ([16]). Some studies have performed cross-country ([17,18]) or cross-city ([19]) comparisons. [20] and [21] consider whether the car is used as first or second family car. [22] differentiates her estimates by residential density. [23] adopt a persona-based approach to represent six diverse driver profiles with different mobility patterns in Flanders. Frequently, authors use sensitivity analyses to explore the impact of changes in the model parameters, including the annual distance traveled assumption on the TCO.

Much fewer contributions analyze the TCO of BEVs in the public sector (unless we consider the urban bus market). We have been able to trace only the following studies searching both the academic and grey literature.

An early study was conducted by [24] on the municipal fleet vehicles' electrification and photovoltaic power in the City of Pittsburgh. Not surprisingly, given the early days of EVs, they find that conventional vehicle would cost less to operate even in a timeframe of more than 15 years. This is due to capital cost involved in purchasing the vehicles and the charging stations as well as to the limited amount of miles these vehicle travel per year (6100 miles). The negative net present value could be partly reduced by accounting for the environmental benefits and the energy production with photovoltaic systems.

Reference [25] compiled a report which aims at identifying ways to accelerate EV adoption by public fleets. Their main proposal is to set up a multi-state EV solicitation to increase the likelihood of favorable terms for acquiring EVs (discounts through volume purchases).

From November 2015 to June 2017 [26] conducted a study to identify the current and forecasted percentage of EVs in the public administration and in the public companies in Hamburg and in its metropolitan region. They interviewed 18 public organizations that operate 72% of the vehicles of all public organizations. They forecast the surveyed public companies will reach a share of 26%–28% EVs until 2020. The number of EVs needed has been calculated assuming that each vehicle accounts for 8000 km per year.

Reference [27] investigates the possibility of introducing electric light trucks for waste collection in Portugal. He argues that they are more efficient, economically reliable and contribute to the reduction of carbon dioxide emissions, especially when used for nocturnal collection of urban waste.

Reference [28] explores the procurement, use and experience of EVs in Danish municipalities using a combination of in-depth surveys and direct interviews. She finds that EVs are most suited to certain departments and that acceptance and uptake of EVs have been complex and not straightforward. She documents that the Danish municipalities were early adopter and procured most of their EVs before 2016. One of the main complaints they face from the users were insecurity about the driving range (especially during the winter season) which made them hesitant about taking it outside the main towns.

## 3. The Public Administrations' Fleet in Friuli Venezia Giulia

### 3.1. Public Administrations in the Friuli Venezia Giulia Region

In an effort to increase policy efficiency and to recognize cultural differences, Italy has decentralized several political and administrative functions to the Regions. Consequently, in every Italian Region such as Friuli Venezia Giulia (a 1.5 million inhabitants Region located in the North East of the Italy), there is a large variety of public entities (authorities, agencies, departments). Some of them are dependent from the national government ministries (e.g., high schools, universities, courts, army, Italian police and Carabinieri, custom agency, heritage protection agencies), while others are under the jurisdiction of the regional Authorities. Among the latter, several public entities own and manage their vehicles.

During the year 2018, we performed face-to-face interviews with the main Region-dependent public entities and collected data on their vehicle fleets, use and needs. Specifically, we interviewed 7 Local Health Authorities, 4 main municipalities (Trieste, Udine, Pordenone and Gorizia) and the 11 sectoral Regional Authorities. A total of 55 small- and medium-sized municipalities send us the data via email (Table 1). Based on the service they offer, we clustered them into three groups: Public Health Authorities, Municipalities and Other Regional Authorities.

**Table 1.** Public Administrations that have formally joined the Noemix project and sent the requested data.

| Public Health Authorities. | Large Municipalities and Medium-Small Municipalities. | Other Regional Authorities. |
|---|---|---|
| AAS2 - Bassa Friulana-Isontina; AAS3 - Alto Friuli; AAS5 - Friuli Occidentale; AsuiTS.; AsuiUD.; IRCCS - Burlo; IRCCS CRO Aviano. | Large municipalities: Gorizia; Pordenone; Trieste; Udine. Medium-small municipalities: Aiello del Friuli; Ampezzo; Artegna; Azzano Decimo; Basiliano; Bertiolo; Buttrio; Caneva; Carlino; Casarsa della Delizia; Castions di Strada; Cavasso Nuovo; Cordenons; Dignano; Fagagna; Flaibano; Forni di Sotto; Gonars; Latisana; Mereto di Tomba; Montereale Valcellina; Moruzzo; Muzzana del Turgnano; Nimis; Palazzolo dello Stella; Pavia di Udine; Pocenia; Porcia; Porpetto; Prata di Pordenone; Precenicco; Premariacco; Pulfero; Remanzacco; Resiutta; Rivignano-Teor; Ronchis; Roveredo in Piano; S. Giovanni al Natisone; Sacile; San Quirino; San Leonardo; San Vito di Fagagna; Sedegliano; Tarcento; Tavagnacco; UTI Carnia; UTI Collinare; UTI Gemonese; UTI Tagliamento; UTI Torre; Valvasone-Arzene; Varmo; Villesse; Zoppola. | Area Science Park; ARPA.; Autorità Portuale; Consorzio Bonifica; Ente tutela patrimonio ittico; ERSA.; FVG Strade; Parco Prealpi Giulie; Regione FVG; Servizio foreste e corpo forestale; Università degli Studi di Trieste. |

The Local Health Authorities are part of publicly funded healthcare system. Each authority provides a comprehensive range of health services, free at the point of use for the resident within a given territory. In Italy, Regions have the financial and administrative responsibility over their Local Health Authorities. The Local Health Authorities have a large fleet of vehicles which are used by nurses for home nursing, social workers for social assistance, doctors, vets, administrative staff, etc., including mostly passenger cars and mixed-use small LCVs but also special purposes vehicles such as ambulances.

Municipalities (4 with more than 40 thousand inhabitants each and 55 with less than 40 thousand inhabitants each) provide administrative, police and social services. They manage also several motorcycles or mopeds, used mostly by the local police, and special purpose vehicles managed by the fire brigade or the gardeners.

The Friuli Venezia Giulia Region has also special purpose public entities (authorities, agencies, universities) devoted to road construction and maintenance, forest and wildlife, water management and fish protection, port management and scientific research and education. Some of them have quite a large fleet of (sometime special) vehicles used to perform their services over the entire region.

We collected data over 3213 vehicles (Table 2). We have classified them into 5 categories according to their function:

1. passenger cars used primarily for people movements, although in some cases they are also useful for transporting light-type work tools, i.e., sanitary appliances, measuring devices, etc.;
2. mixed-use small LCVs used both for people movements and to transport light work tools;
3. large LCVs or heavy duty vehicles (HDVs) such as trucks, vans, buses, coaches used for freight transport or for carrying a large number of people;
4. motorcycles and mopeds;
5. special purpose vehicles: e.g., ambulances or other vehicles for special use (advanced rescue, transport of animals).

Passenger cars represent 57.2% of the total and 15.9% are mixed-use small LCVs. For the purpose of this paper, we will focus only on the first two categories, hence on a sample of 2349 (73.1%) vehicles since, at the present state of maturity of the electric vehicle market, there exists a technically suitable supply of comparable electric vehicles only for these two car segments. Local Health Authorities manage more than half of the passenger cars of our sample, while the mixed-use small LCVs are more equally distributed among the three groups.

**Table 2.** Vehicles in the three public sector groups.

| Public Sector Entities | N° | Passenger Cars | Mixed-Use Small LCVs | Large LCVs or HDVs | Motorcyclesand Mopeds | Special Purpose Vehicles | Total |
|---|---|---|---|---|---|---|---|
| Local Health Authorities | 7 | 946 | 184 | 98 | 20 | 83 | 1331 |
| Municipalities | 59 | 577 | 141 | 267 | 102 | 127 | 1214 |
| Other Regional Authorities | 11 | 315 | 186 | 111 | 0 | 56 | 668 |
| Total | 77 | 1838 | 511 | 476 | 122 | 266 | 3213 |

*3.2. Main Characteristics of the Public Sector Fleets*

The average age of the passenger and mixed-use vehicles in the fleets is 11.5 years old. The age distribution is reported in Figure 1. A total of 22% of the vehicles are at least 15 years old, more than half are at least 10 years old and only 11% have been bought less than 4 years ago. This finding demonstrates how the budget constraints are limiting the substitution possibilities of the fleet managers. The interviews and the financial data we collected indicated that because of their age most of the cars required heavy maintenance costs. A further implication is that 74% of them have an engine technology that does not respect the Euro 5 emission standards. The current ICEV fleet has, hence, a quite low air pollution efficiency.

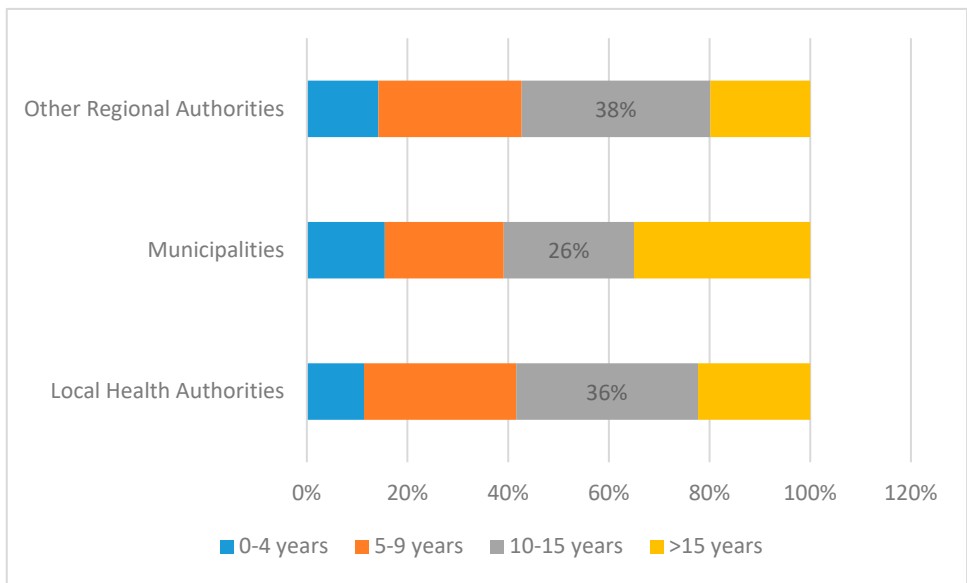

**Figure 1.** Age distribution of the vehicles.

In terms of brand, Fiat (FCA) is by far the most represented brand (87%), with its various models but mostly with Fiat Panda, Fiat Punto and Fiat Doblò for the mixed-use small LCVs. Renault follows at a large distance (5%), mostly with the Renault Clio. In fact, a specific characteristic of the FVG public sector fleet is the large proportion of small vehicles, belonging to the segments A and B. In terms of fuel, about 80.3% run on petrol (including a small share of petrol/LPG fuel vehicles equal to 2.5%, and 0.8% of petrol/methane ones), 18.2% are diesel-fueled and only 1.5% are electric cars. The latter are 15 electric cars owned by the Municipality of Udine as part of a pilot trial electric carsharing service.

Concerning the annual distance traveled (ADT), most vehicles (61.4%) travel up to 10,000 km annually, just over one-fifth are driven annually between 10,000 and 15,000 km and 16.9% distances larger than 15,000 km (Figures 2 and 3). Unfortunately, fleet managers were not able to inform us on the daily distances traveled by each car. Dividing the annual mileage of each vehicle by the number of working days in a year, assumed to be equal to 270, we estimate that 40.3% of the vehicles travel less than 25 km a day and only 3.8% more than 100 km. We double-check our estimates with the paper records that some agencies have on the daily vehicle use and we can confidently argue that our estimate is correct. Fleet managers agree with our conclusions as well. Consequently, we feel safe to state that in most of the cases the EVs currently available in the market are able to satisfy the daily travel needs of the interviewed public sector entities. This is an important finding since early adopter such as the Danish Municipalities complained about the limited driving range especially in the winter season [28].

A further information from Figures 2 and 3 is that the public entities classified as Other Regional Authorities have a larger share of vehicles driving longer distances since their responsibility to serve the entire FVG Region. Municipalities serve the areas within their administrative boundaries, hence, they have a high percentage of small distance trips, so that more than 50% of the vehicles have an ADT lower than 7500. Local Health Authorities serve several contiguous municipalities; hence, their vehicles cover distances intermediate between the two previous groups. The ADT variable, as explained below, has a very large impact on the share of vehicles that is economically motivated to be electric.

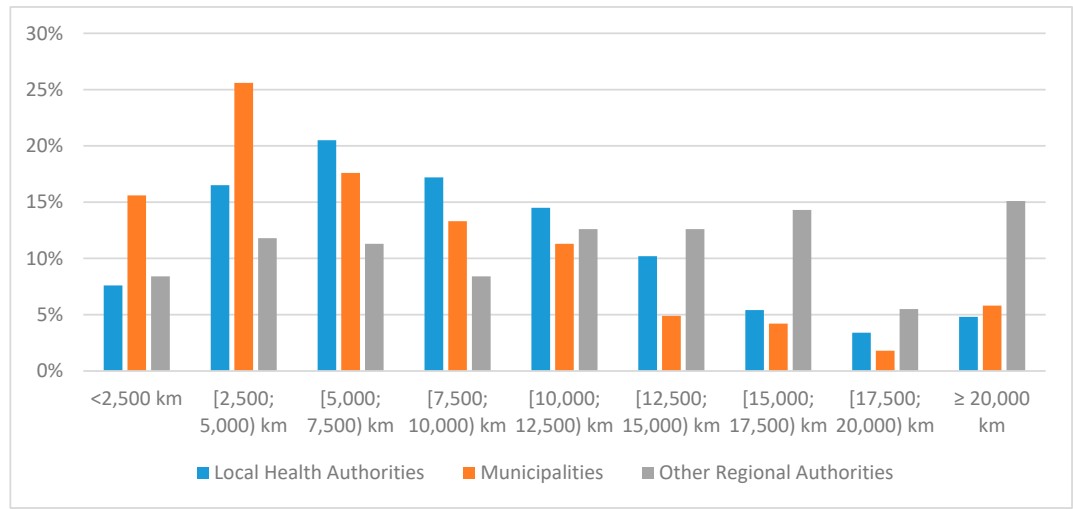

**Figure 2.** Percentage of passenger cars in each annual distance traveled (ADT) (km) class (The interviewed Public Entities did not respond with the same degree of detail to the requests for information, therefore our analyses are based, for each aspect investigated, on the available data).

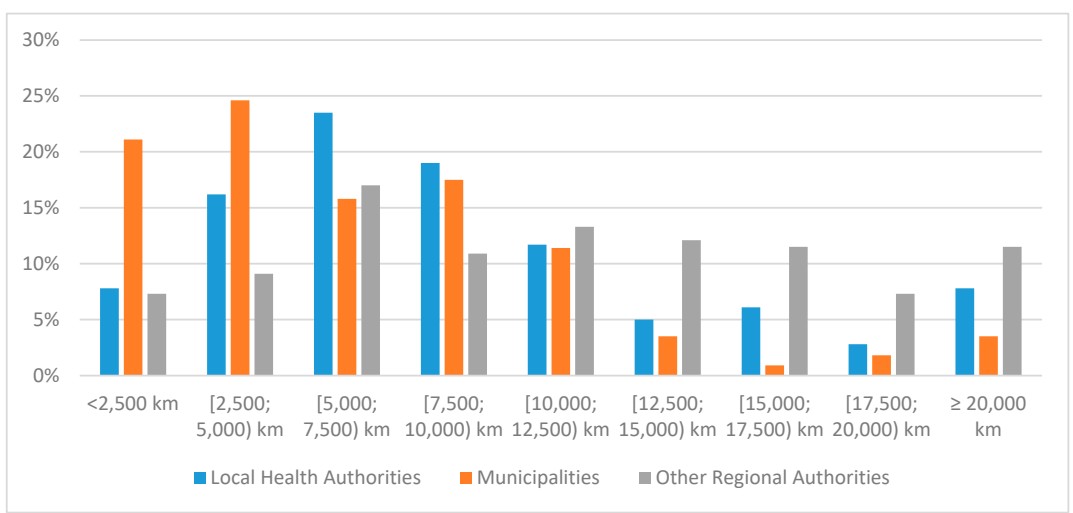

**Figure 3.** Percentage of mixed-use light commercial vehicles (LCV) in each ADT (km) class.

## 4. Total Cost of Ownership Model

The total cost of ownership (purchasing and operating) of a vehicle for a given year per km driven (TCO/km) can be estimated with the following formula (for more details on the TCO model, see [9,13,16]):

$$\frac{\text{TCO}}{\text{km}} = \frac{(\text{IC} - \text{RV} \cdot \text{PVF}) \cdot \text{CRF} + \frac{1}{T} \sum_{t=1}^{T} \frac{\text{AOC}_t}{(1+i)^t}}{\text{ADT}} \tag{1}$$

where IC are the initial costs equal to:

$$\text{IC} = \text{MSRP} - \text{RD} + \text{RC} - \text{SUB} + \text{INFRA} \tag{2}$$

MSRP is the manufacturer's suggested retail price, RD is the potential retailer's discounts, RC are registration costs, SUB is the government subsidies and INFRA is the charging infrastructure costs. In Italy, the government subsidized EVs up to €6000 for private buyers only. Hence, SUB will be set to zero in our estimate. To the initial costs, one must add the annual operating costs (AOC) during the period of use.

$$\text{AOC} = \frac{1}{T}\sum_{t=1}^{T}\frac{\text{AOC}_t}{(1+i)^t} = \frac{1}{T}\sum_{t=1}^{T}\frac{\text{INS}_t + \text{MAINT}_t + F_t/E_t}{(1+i)^t} = \frac{1}{T}\sum_{t=1}^{T}\frac{\text{INS}_t + \text{MAINT}_t + \text{WAEP}\cdot\gamma\cdot(\alpha\cdot\text{FE}_{\text{urb}} + (1-\alpha)\cdot\text{FE}_{\text{exturb}})}{(1+i)^t} \quad (3)$$

AOC is computed as the discounted average of all costs incurred during the period of ownership T of the vehicle. These costs include the annual insurance premium, maintenance costs ($\text{MAINT}_t$) and the costs of fuel/electricity ($F_t/E_t$). The latter are quite difficult to estimate and cause a certain degree of uncertainty. In fact, fuel/energy efficiency depends on many factors, related to traffic situations (congestion levels), road type (flat or steep), driving style and the urban ($\text{FE}_{\text{urb}}$) or extraurban ($\text{FE}_{\text{exturb}}$) nature of trips

In the case of EVs, charging cost might be very different depending on the charging station used (free, domestic, public), charging speed (in time, slow, fast) and the time of the day when charging takes place. The former is captured by the variable WAEP that is the weighted average of the prices prevailing by using different charging infrastructures. In addition, there is the seasonal factor $\gamma$, since EVs efficiency is affected by extreme weather conditions. Furthermore, we denote $\alpha$ to be the percentage of trips made in an urban area, accounting for the fact EVs perform relatively better in urban traffic than in intercity driving.

A third cost component is the loss of value of the vehicle. This is included in our formula as residual value (RV) of the vehicle to be subtracted at time T, when the vehicle is sold or disposed of, after been properly discounted (PVF). Since the first part of the numerator in Equation (1) are lump-sum costs while AOC occurs at an annual basis, the former needs to be multiplied by the CRF (The capital recovery factor is the amount of equal (or uniform) payments to be received for T years such that the total present value of all these equal payments is equivalent to a payment of one euro at present, if interest rate is i). i.e., the capital recovery factor equal to $(i(1+i)^T)/((1+i)^T - 1)$, to obtain the average annual fixed costs of owning the vehicle. Hence, the numerator represents the annualized total cost of ownership (ATCO). Dividing it by the annual distance travelled (ADT), one obtains an estimate of the TCO/km. Such metric is used to compare cars with alternative propulsion systems.

## 5. Vehicles Considered and Model Parameters

The model has been estimated for 26 ICEVs (5 passenger cars and 21 LCVs) and 11 BEVs (6 passenger cars and 5 LCVs). The electric passenger cars have been selected among the ones available in Italy in 2020 in the segments A and B (with the exception of the Nissan Leaf, which belongs to the segment C). ICE passenger cars have been selected considering the one mostly used in the public sector fleets. A similar criterion has been applied for the selection of the LCVs, with a focus mostly on the small LCVs. In this segment, there is a large variety of ICE models and brands, both diesel- and petrol-fueled, but only a limited number of electric ones.

The starting point is the estimation of the baseline scenario, based on the parameters reported in Table 3 They have been selected on the basis on the information collected during the interviews for ICEVs and that derived from the desktop research for the BEVs. Some of these assumptions will be modified in the scenario analyses.

**Table 3.** Model parameter assumptions in the baseline scenario.

| Initial costs: | Annual Operating Costs: |
|---|---|
| • MSRP: see Table 3. <br><br>     • Discount on purchasing price for BEVs: 0 <br>     • Charging infrastructure and installation costs per vehicle: 1000 € <br>     • Resale value after 10 years: 20% of MSRP for ICEVs and 10% for EVs | • Petrol price: 1.59 €/liter <br> • Diesel price: 1.48 €/liter <br> • Maintenance: for EVs 50% of ICEVs <br> • Insurance: derived from on line quotes |
| **Ownership and financial parameters:** <br><br> • Ownership period: 10 years <br> • Interest rate: 2% | **Travel habits:** <br><br> • Annual distance travelled in km (ADT): 10,000 km <br> • % of urban travel: 80% |
| **Charging habits:** <br><br> • Electricity price when charging at its own premise: 0.18 €/kWh <br> • Electricity price when charging at public stations: 0.50 €/kWh <br> • Frequency of charges at its own premise: 90% | • Social costs: not included |

## 6. Results and Scenario Analyses

We report the results of three scenario analyses:

- a baseline Scenario A, estimated initially with the parameters illustrated in Table 2 and then for the different ADT classes;
- a social cost internalization Scenario B, which incorporates a social cost estimate of the air pollution generated by ICEVs and saved with BEVs;
- a discounted-MSRP Scenario C which assumes that the public sector can obtain their BEVs at discounted prices from the car manufacturers;
- a socially optimal and discounted Scenario D that assumes the simultaneous occurrence of scenarios B and C.

The aim is to estimate what percentage of the public sector fleet would make economically sense to turn into electric.

Table 4 illustrates our estimates for each passenger car in the baseline scenario (ADT = 10,000 km) and reports in the last rows the average values for BEVs and petrol fueled ICEVs. The main finding is that EVs are on average not cost-competitive: their average TCO/km is higher than the petrol cars one (€0.37 vs. €0.31). This result is due to the much higher average MSRP of the BEVs (more than €10,000 higher) which is not compensated by the annual operating cost savings (about €800), even after 10 years of use at 10,000 km per year. Note also that the Fiat Panda has the lowest TCO/km among all passenger cars, and the Skoda Citigo e-iV among the BEVs.

To account for the fact that vehicles have different ADTs as illustrated in Figures 2 and 3, we calculate and report in Table 5, for each distance class, the average distance within each class and estimate the corresponding TCO/km. In the case of Local Health authorities, for instance, the passenger cars belonging to the 0–2500 km distance class travel on average 1244 km per year. Similarly, we estimate the average distance for each distance class, which might differ among the three groups of agencies. We use these average ADTs in the TCO model to estimate the corresponding TCO/km for the average BEV and the average ICEV.

By average BEV, we mean a BEV having the average cost structure calculated on the basis of the vehicle models considered. A similar interpretation applies for the average petrol car. Subsequently,

we estimate the TCO/km for each distance class and report the results in Table 5. They can be interpreted as the results of the Status Quo Scenario A.

We find that in the lowest distance class (i.e., on average 1244 km per year), an average BEV would have a TCO/km equal to €3.30, which would be much higher than that of an average petrol fueled car (equal to €1.80). Reading the results for the various ADT classes, one can see that only in the highest ADT class (on average 29,934 km per year), a BEV would have a TCO/km lower than an ICEV (In performing our calculations, we keep the 10-years ownership period constant. As pointed out by an anonymous reviewer, higher ADTs increase battery degradation and might cause the need to replace the battery. Currently, car manufacturers issue an 8 years/160,000 km warranty on the battery. A common interpretation is that they feel reassured that the battery will normally reach that longevity/range. In our analysis, the assumption is that this threshold could be overcome. There is anecdotal evidence that this is the case. However, very few EVs have reached that longevity/range to allow us to state with a large degree of certainty that no battery substitution would be needed. This represents a caveat in the interpretation of our results). When that happens, we wrote in bold the percentage of passenger cars that belong to the class. In summary, we find that in the case of the Local Health Authorities 4.8% of the passenger cars would be cost competitive if they were BEVs instead of ICEVs. In the case of Other Regional Agencies, the percentage increases to 15.1% of the passenger cars in the fleet, while in the case of Municipalities BEVs are never cost competitive.

**Table 4.** Estimates for passenger cars: baseline scenario.

| Passenger Car Model | Fuel | MSRP (€) | Annualized Initial Costs (AIC) (€) | Annual Operating Costs (AOC) (€) | Annualized TCO (ATCO) (€) | TCO/km (€) |
|---|---|---|---|---|---|---|
| Renault Clio | Petrol | 13,350 | 1166 | 1631 | 2796 | 0.28 |
| Fiat 500 | Petrol | 14,250 | 1241 | 1631 | 2872 | 0.29 |
| Fiat Panda | Petrol | 11,600 | 1018 | 1631 | 2649 | 0.26 |
| Nissan Pulsar | Petrol | 23,640 | 2046 | 2299 | 4345 | 0.43 |
| Smart | Petrol | 14,034 | 1218 | 1631 | 2848 | 0.28 |
| Nissan Leaf Acenta 3.6 | BEV | 30,700 | 3154 | 1099 | 4254 | 0.43 |
| VW e-UP | BEV | 23,350 | 2436 | 870 | 3306 | 0.33 |
| Skoda Citigo e-iV | BEV | 22,300 | 2333 | 906 | 3239 | 0.32 |
| Seat Mii | BEV | 22,500 | 2353 | 870 | 3223 | 0.32 |
| Smart EQ | BEV | 24,559 | 2542 | 996 | 3538 | 0.35 |
| Renault Zoe R110 Life | BEV | 34,100 | 3476 | 978 | 4454 | 0.45 |
| Average | BEV | 26,252 | 2716 | 953 | 3669 | 0.37 |
| Average | Petrol | 15,375 | 1338 | 1764 | 3102 | 0.31 |

**Table 5.** TCO estimates for passenger cars with varying ADT: Status quo scenario A.

| Local Health Authorities | | | | Municipalities | | | | Other Regional Agencies | | | |
|---|---|---|---|---|---|---|---|---|---|---|---|
| ADT (km) | BEV | ICEV | % | ADT (km) | BEV | ICEV | % | ADT (km) | BEV | ICEV | % |
| 1244 | 3.30 | 1.80 | 7.6 | 1459 | 3.05 | 1.67 | 15.6 | 1017 | 3.10 | 1.74 | 8.4 |
| 3927 | 1.12 | 0.68 | 16.5 | 3673 | 1.29 | 0.77 | 25.6 | 3696 | 0.90 | 0.56 | 11.8 |
| 6196 | 0.73 | 0.48 | 20.5 | 5993 | 0.77 | 0.50 | 17.6 | 6284 | 0.55 | 0.38 | 11.3 |
| 8723 | 0.55 | 0.39 | 17.2 | 8465 | 0.57 | 0.40 | 13.3 | 8657 | 0.42 | 0.30 | 8.4 |
| 11,200 | 0.44 | 0.33 | 14.5 | 10,757 | 0.46 | 0.34 | 11.3 | 11,179 | 0.33 | 0.26 | 12.6 |
| 13,695 | 0.37 | 0.30 | 10.2 | 13,526 | 0.38 | 0.30 | 4.9 | 13,910 | 0.28 | 0.23 | 12.6 |
| 16,116 | 0.32 | 0.27 | 5.4 | 15,568 | 0.34 | 0.28 | 4.2 | 16,001 | 0.25 | 0.22 | 14.3 |
| 18,671 | 0.29 | 0.26 | 3.4 | 18,647 | 0.29 | 0.26 | 1.8 | 18,662 | 0.22 | 0.20 | 5.5 |
| 29,934 | 0.19 | 0.20 | **4.8** | 24,350 | 0.24 | 0.23 | 5.8 | 30,385 | 0.16 | 0.17 | **15.1** |

An alternative to the comparison based on the average car costs (BEV and petrol car) is to adopt the cost structure of the least expensive BEV (the Skoda Citigo e-iV) and of the least expensive petrol car (Fiat Panda). The idea is that the fleet managers decide what car to buy looking for the car model

that has the minimum cost for each propulsion system. Performing the same exercise, we obtain the results presented in the second row of Table 6. The two criteria lead to the same result for the Local Health Authorities and the Other Regional Authorities, while the second criterion leads Municipalities to substitute 5.8% of their passenger car fleet with BEVs. Table 6 reports also the ADT over which BEVs are cost competitive with ICEV. It can be notice that they correspond to high ADT, corresponding to daily distance of over 100 km.

**Table 6.** Percentage of passenger cars and mixed-use small LCVs that if battery electric vehicles (BEVs) would have a lower TCO/km: Scenario A status quo.

| | | Local Health Authorities | | Municipalities | | Other Regional Agencies | | Ave-Rage |
|---|---|---|---|---|---|---|---|---|
| | Criterion | Min. ADT | % | Min. ADT | % | Min. ADT | % | % |
| Passenger cars | Av. TCO/km | 29,934 | 4.8 | | 0 | 30,385 | 15.1 | 5.1% |
| | Min. TCO/km | 29,934 | 4.8 | 24,350 | 5.8 | 30,385 | 15.1 | 6.9% |
| Mixed-use small LCVs | Av. TCO/km | 32,906 | 7.8 | | 0 | | 0 | 2.8% |
| | Min. TCO/km | 32,906 | 7.8 | | 0 | | 0 | 2.8% |

The same calculations are made for LCVs. The detailed results for each mixed-use LCV are reported in Table 7. By applying the two criteria to the LCVs, we obtain the results reported in the third and fourth row of Table 6. They lead to the same result. Namely, only in the case of the Local Health Authorities electric LCVs are cost competitive for the highest ADT class comprising 7.8% of their entire fleet. The ADT requirement for electric mixed-use small LCVs is 32,906 km per year, a challenging requirement for the current electric LCVs available in the market. Differently from the passenger cars, in this case a charging session during the daily use might be needed.

**Table 7.** Estimates for mixed-use LCVs: baseline scenario.

| Mixed-use LCV | Fuel | MSRP (€) | Annualized Initial Costs (AIC) (€) | Annual Operating Costs (AOC) (€) | Annualized TCO (ATCO) (€) | TCO/km (€) |
|---|---|---|---|---|---|---|
| Mercedes Citan 1.5 109 Cdi S&S Compact | diesel | 15,316 | 1468 | 1813 | 3281 | 0.33 |
| Renault Kangoo 1.5dci 90 EDC Express Ice | diesel | 18,300 | 1746 | 1829 | 3574 | 0.36 |
| Peugeot Partner Blue Hdi 75 PC Pro | diesel | 14,380 | 1381 | 1806 | 3187 | 0.32 |
| Peugeot Bipper 1.3 Hdi 80CV Pro | diesel | 13,100 | 1262 | 1750 | 3012 | 0.30 |
| Ford Transit Courier 1.5 Tdci 100CV Van Trend | diesel | 15,000 | 1438 | 1762 | 3200 | 0.32 |
| Fiat Fiorino 1.3 Mjt 80cv Cargo | diesel | 14,190 | 1363 | 1903 | 3266 | 0.33 |
| Fiat Doblo' 1.6 Mjt 105cv Pc-Tn Cargo Lam. Sx | diesel | 19,150 | 1825 | 1920 | 3745 | 0.37 |
| Citroen Berlingo Blue hdi 75 Van M Club | diesel | 16,380 | 1567 | 1806 | 3373 | 0.34 |
| Vw Caddy 2.0 Tdi 122 Cv 4mot. | diesel | 21,180 | 2014 | 2085 | 4098 | 0.41 |
| Nissan NV200 1.5 Dci 90CV | diesel | 16,331 | 1562 | 1877 | 3439 | 0.34 |
| Nissan Nv200 1.6 | petrol | 14,331 | 1376 | 2358 | 3734 | 0.37 |
| Mercedes Citan 1.2 112 S&S Long | petrol | 15,138 | 1451 | 2238 | 3689 | 0.37 |
| Ford Transit Courier 1.0 Ecob.100CV Van Trend | petrol | 13,000 | 1252 | 2003 | 3255 | 0.33 |
| Fiat Doblo' 1.4 Pc-Tn Cargo Sx | petrol | 16,550 | 1583 | 2421 | 4004 | 0.40 |
| Fiat Fiorino 1.4 8v 77cv Cargo | petrol | 12,690 | 1223 | 2328 | 3552 | 0.36 |
| Renault Kangoo 1.2tce 115 EDC Express Ice | BEV | 17,600 | 1680 | 2321 | 4002 | 0.40 |
| Nissan E-NV200 40kWh Visia Van | BEV | 43,200 | 4569 | 1318 | 5886 | 0.59 |
| Renault Kangoo Z.E. Ice Flex 4p. 44 kWh | BEV | 27,650 | 2980 | 1285 | 4265 | 0.43 |
| Peugeot Partner Tepee Full Electric Active | BEV | 33,000 | 3526 | 1140 | 4666 | 0.47 |
| Citroën E-Berlingo Multispace | BEV | 33,000 | 3526 | 1140 | 4666 | 0.47 |
| Average | ICEV | 15,790 | 1512 | 2014 | 3526 | 0.35 |
| Average | BEV | 34,213 | 3650 | 1221 | 4871 | 0.49 |

The regional entities, however, are interested not only in the private cost comparison but also in the social benefits of reducing air pollution and noise. In the Scenario B, we account for such benefits by incorporating in the TCO/km comparison the social cost per km of driving an ICEV versus a BEV for

the FVG Region. From a purely monetary TCO analysis we turn to a society-oriented TCO analysis [4]. Accounting for the social costs/benefits, however, it is not an easy task. One should consider the air and noise emissions generated, the location where it takes place, how the electricity is produced and what are the corresponding economic values of the damages caused on the local population and on the environment of each propulsion system. In this paper, we will not discuss in detail these topics. Instead, we rely on a previous report prepared to set the guidelines for planning the EV charging infrastructure in the FVG Region, where we estimated the social cost difference between an ICEV and a BEV to be equal to 0.02 €/km, comprising $CO_2$, local air pollution and noise emissions [29] (The FVG region is characterized by a cluster of several small-to-medium sized cities and towns, with only the cities of Trieste and Udine having more than 100 thousand inhabitants. Taking into account the traffic characteristics and using the HEATCO parameters ([30]), [29] estimates that 5% of the total social cost of an ICE vehicle is attributable to $CO_2$ emissions, 25% to local air pollution and 70% to traffic noise). The social benefits' evaluation is based on a meta-analysis published in the European project HEATCO [30] and on the traffic characteristics of the FVG cities. We will adopt this figure in Scenario B.

Scenario C assumes a different perspective: it takes into account the possibility that the public sector might obtain a discount when procuring EVs in high volume, along the lines suggested by [25]. We assume a discount equal to €2000. Such a potential discount has two motivations: a) the procurement of a large number of EVs by the Region FVG instead of from every single entity, and b) a marketing motivation for the vehicle manufactures, since the public sector represents a showcase for demonstrating the advantages of specific EV brand/model. We fixed the discount at a conservative level, less than 10% of the current MSRP, after informal discussions with representatives of the car dealers. Currently, there is no empirical evidence on real-world figures, which tend to be kept unofficial for business privacy motivations.

Scenario D assumes the joint realization of Scenarios B and C.

Table 8 reports the percentage of passenger cars and LCVs that would make economic sense to substitute under the various scenarios and considering the two valuation criteria (average and minimum costs). We illustrate the results for each public sector group and the minimum ADT that make BEVs competitive. For instance, in Scenario B 4.8% of the passenger cars of the Local Health Authorities should be electric, that is the ones which travel more than 29,934 km per year; 5.8% of the those of the Municipalities (those driven more than 24,350 km per year); and 15.1% of the those of the Municipalities (driven more than 30,385 km per year). Although the ADT requirements are high, they amount to not much more than 100 km per day, which is a range that most of the recent EVs can deliver. On average, the percentage is 6.9% when the average TCO/km criterion is used to select which car to buy. When the minimum TCO/km criterion is used, the average percentage suitable for a switch to BEVs becomes 16.7% and the ADT requirements drop considerably.

For mixed-use small LCVs, the percentage that make economic sense to turn to electric is on average equal to 8%, irrespective of the criterion. The ADT requirements are high a just within the driving range limits of the current electric LCVs on sale in Italy. Hence, a driving range limitation might still be a concern in this market segment.

It can also be noted that the percentage varies by type of agency. It is lower for Municipalities that have a more limited territorial scope for their services, intermediate for the Local Health Authorities and higher for Other Regional Authorities, which serve with their cars the entire region.

Scenario C determines an EV percentage slightly lower than that obtained considering the social cost internalization scenario B. ADT requirements remain high in both cases. Only the joint application of Scenarios B and C generates an average percentage of substitution with electric passenger cars much higher, between 10.1% and 25.6%. The percentage increase considerably especially for the Other Regional Agencies, and the ADT requirements drop to below 14,000 km per year.

In the case of the mixed-use small LCVs, however, the percentage which would be justified to turn into electric vehicles on economy grounds remains small (on average 8%–9.5% of the fleet), with still high ADT requirements.

**Table 8.** Percentage of passenger cars and mixed-use small LCVs that if BEVs would have a lower TCO/km.

| | Criterion | Local Health Authorities | | Municipalities | | Other Regional Agencies | | Ave-Rage |
|---|---|---|---|---|---|---|---|---|
| | | Min. ADT | % | Min. ADT | % | Min. ADT | % | % |
| Scenario B: social cost internalization | | | | | | | | |
| Passenger cars | Av. TCO/km | 29,934 | 4.8 | 24,350 | 5.8 | 30,385 | 15.1 | 6.9 |
| | Min. TCO/km | 16,116 | 13.6 | 15,568 | 11.8 | 16,001 | 34.9 | 16.7 |
| Mixed-use small LCVs | Av. TCO/km | 32,906 | 7.8 | 23,073 | 3.5 | 25,484 | 11.5 | 8.0 |
| | Min. TCO/km | 32,906 | 7.8 | 23,073 | 3.5 | 25,484 | 11.5 | 8.0 |
| Scenario C: €2,000 volume discount | | | | | | | | |
| Passenger cars | Av. TCO/km | 29,934 | 4.8 | 24,350 | 5.8 | 30,385 | 15.1 | 6.9 |
| | Min. TCO/km | 18,671 | 8.2 | 18,647 | 7.6 | 18,662 | 20.6 | 10.1 |
| Mixed-use small LCVs | Av. TCO/km | 32,906 | 7.8 | | 0.0 | 25,484 | 11.5 | 7.0 |
| | Min. TCO/km | 32,906 | 7.8 | | 0.0 | 25,484 | 11.5 | 7.0 |
| Scenario D: social cost internalization and €2,000 volume discount | | | | | | | | |
| Passenger cars | Av. TCO/km | 18,671 | 8.2 | 18,647 | 7.6 | 18,662 | 20.6 | 10.1 |
| | Min. TCO/km | 13,695 | 23.8 | 13,526 | 16.7 | 13,910 | 47.5 | 25.6 |
| Mixed-use small LCVs | Av. TCO/km | 32,906 | 7.8 | 23,073 | 3.5 | 25,484 | 11.5 | 8.0 |
| | Min. TCO/km | 18,327 | 10.6 | 18,298 | 5.3 | 25,484 | 11.5 | 9.5 |

## 7. Discussion and Conclusion

Public sectors fleet managers have the difficult task of making decisions between conflicting goals such as using efficiently the taxpayers' money and "greening" their fleets. The FVG Region embraced this task and, within the H2020 European project Noemix, explored the technical and economic rationale of substituting its current vehicles with electric ones. The survey on the existing fleet, mobility demand and fuel/energy efficiency and prices allowed us to estimate the metric TCO/km for a large number of vehicles in use in the FVG public sector fleets and to compare it with that of their EV counterparts. We performed estimates differentiating among three groups of public entities (Local Health Authorities, Municipalities and Other Regional Agencies) and between passenger cars and mixed-use small LCVs.

An encouraging finding is that the mobility needs of most of the public entities are well within the driving range of the EVs currently in the market, without requiring charging during the daily service. The fears and the difficulties of the early adopters (e.g., the Danish Municipalities) seem to be overcome by the increase in the driving range of the recent EVs, which are available in the market at prices similar to those prevailing in 2016. However, such a positive factor turns into a negative one from an economic point of view, since the limited ADTs of large part of the vehicles in the fleet do not justify their acquisition.

However, the results vary by scenario. Under the Scenario A Status Quo (current cost structure and only private costs), we find that it would make economic sense to substitute with EVs only a rather small percentage of the current passenger cars (5.1–6.9%) and an even smaller percentage of mixed-use small LCVs (2.8%). This finding reflects the fact that the MSRP of EVs is still much higher than that of the ICEVs, and that such price difference is not compensated by the savings in the annual operating costs, even after 10 years.

In the social cost internalization scenario (Scenario B), the percentage increases to 6.9–16.7% for the passenger cars and to 8% for the mixed-use small LCVs. An almost equivalent increase might

be reached if the public sector is able to obtain a €2000 MSRP volume discount from the vehicle manufactures (Scenario C). The combination of the previous two scenarios (internalization + discount) indicates an economically motivated substitution of 10.1–25.6% of the passenger cars and of 8.0–9.5% of the mixed-use small LCVs.

Consequently, overall we find positive although relatively small percentages. Important determinants of this finding are, in our view, the still high battery costs and the still limited supply of vehicles in the segments A and B for passenger cars and, even more limited for small LCVs. In addition, in Italy the main national manufacturer, FCA, has not yet produced compelling electric counterparts of the successful models FIAT Panda, FIAT Punto and FIAT 500 (widely adopted in the public sector fleets) or the Fiat Fiorino, Doblò or Ducato in the LCVs segment. There is news that within this year FCA, who reluctantly invested in EVs, is going to bring to the market small electric cars whose technical and economic specifications are, at the time of this writing, not yet available.

Our estimates, however, should encourage fleet managers to gradually start greening the public sector fleet, and the leading role provided by the FVG regional government also with the Noemix project seems worth pursuing. As BEVs increase their uptake, the parameters of the TCO could be further updated and verified, and the methodology we presented in this paper could provide a guide to take economically sound procurement decisions.

Additionally, a potential tool that public sector managers could use to make the introduction of EVs in their fleet economically more justified is the increase of their ADT. Based on our interviews with the public sector fleet managers, these ADT increases are possible but they require organizational and technological innovations to the current management and use practices.

A final caveat concerns the estimate of the marginal social cost gap between ICEVs and BEVs, which, by its nature, contains a high degree of uncertainty and is time- and location-specific [31].

**Author Contributions:** Conceptualization: R.D., M.S., M.G., S.A.; data curation: R.D., M.S.; formal analysis: R.D., M.S.; methodology: R.D., M.S., M.G., S.A.; writing—original draft: R.D., M.S.; writing—review and editing: R.D., M.S., M.G., S.A. All authors have read and agreed to the published version of the manuscript.

**Funding:** The research leading to these results has received funding from the European Union's Horizon 2020 research and innovation programme under grant agreement N° 754145.

**Acknowledgments:** The opinions expressed in the paper reflect only the authors' view. The European Union is not liable for any use that may be made of the information contained therein.

**Conflicts of Interest:** The authors declare no conflict of interest.

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
