# Peer review of "The Economic Case for Electric Vehicles in Public Sector Fleets: An Italian Case Study"

_wevj, doi:10.3390/wevj11010022_

Round 1
Reviewer 1 Report
This paper elaborates on the use of electric vehicles in public sector fleets. Although the paper is clear and consistent, some concerns still exist for this reviewer which needs to be addressed by the authors before publication. My suggestions are listed below.
- Major clarifications and explanations are needed make the contributions and novelties of the paper clearly stand out, and more empirical studies are needed to demonstrate the advantages of the idea proposed. The paper does a good job in describing the method, but since it does not adequately review similar studies fails to demonstrate the novelty and utility of the approach. There exist many works analyzing participation of Electric Vehicles in different sectors especially in energy-related domains which emphasize the role of EVs in future smart societies and greener transition which is helpful to the readers better understand the subject matter. The following could be included in review literature section (https://doi.org/10.1016/j.jclepro.2019.118076, https://doi.org/10.3390/en11092413, DOI: 10.1109/ACCESS.2018.2878903 and many more)
- References to the websites must be provided in reference list according to the standard citation format. Also there are many citation errors in the text “Error! Reference source not found” which have to be fixed!
- Please detail what is the practical outcome of the application. How would it be used by a system operator or any other interested stakeholder/operator?
Reviewer 2 Report
This is an interesting case study to address economic analysis of EVs in public sector fleets in Italy. It is generally well organised and easy to follow. Several important issues are provided for the authors:
- Literature survey is not comprehensive and organised. More relevant journal papers should be addressed and logically elaborated.
- From the methodology side, what are the key difference between the paper and previous studies?
- 2-3 good figures should added in addition to the bunch of tables.
- Several ERROR, Reference not found could be targeted in the paper, which should be carefully eliminated and overall proofreading is in need.
Reviewer 3 Report
The authors present an interesting analysis to evaluate the economic advantages of using EVs for public car fleets in an Italian region. I think the paper is interesting and well-written, and could be useful for the readers of WEVJ. Nonetheless, I think there are some issues that the authors should be clarify before this work can be accepted for publication.
Main issues:
While the authors perform a literature review on the analysis of EVs fleets in the public section, a brief discussion of the results of TCO analysis for private EVs may be adequate, to allow the readers contextualizing their results.
There are dozens of occurrences of (Error! Reference source not found.) in the texts (e.g lines 117, 139, 157,…:).. Please check the pdf version (bibliography and cross-references) before submitting.
Table 7: How are the authors varying the ownership time based on the ADT of the vehicles? Do they consider the same 10-years duration or do the account for the fact that a much higher ADT will reduce the ownership of the vehicles and vice-versa? In particular, do they consider the need of replacing the batteries during the ownership time?
Lines 336-338: The authors should at least describe the role of the different impacts in the 0.02 €/km that they are considering, since the readers should have a basic idea of these aspects without the need of referencing to an external publication (which seems to be in Italian).
Minor issues:
Lines 38-39: The use of electric vehicles does not automatically lead to an “increase of renewables in the electricity mix”, which requires other actions. Please clarify if such actions where included into the project.
Table 1 should be included as an annex.
Lines 151-152: Some municipalities are using electric bus, which seems to have been excluded for the unavailability of competitive solutions. Please justify this choice.
Table 5: “KWh” should be corrected to “kWh”.
Line 333: “In this paper, we will discuss in detail these topics”. It seems that the authors are actually doing the very opposite.
Round 2
Reviewer 1 Report
Authors have satisfactorily addressed my comments and the revised manuscript has been improved greatly.
Reviewer 3 Report
The authors have addressed all the issues I have raised.